# Crystallographic fragment screening supports tool compound discovery and reveals conformational flexibility in human deoxyhypusine synthase

Piotr Wilk [1], Elżbieta Wątor-Wilk [2], Damian Muszak [3], Paweł Kochanowski[1,4], Tobias Krojer [5] & Przemysław Grudnik [1] ✉

Deoxyhypusine synthase (DHS) catalyzes the rate-limiting step of hypusination, a unique post-translational modification of eukaryotic translation factor 5 A (eIF5A). While DHS activity plays a critical role in both normal cellular processes and disease development, the lack of specific molecular tools has hindered detailed studies of this enzyme and the hypusination pathway in general. Existing inhibitors, such as polyamine analogs, suffer from limited specificity and versatility. In this study, we utilized crystallographic fragment screening (CFS) to identify potential DHS inhibitors and explore novel applications of this approach. With an unprecedented hit rate of 39%, we identified fragment clusters binding at key sites, including the active site entrance, the tetramer interface, the regulatory ball-and-chain motif, and potentially allosteric regions on the enzyme's surface. Notably, we discovered a covalent modifier that targets the catalytic lysine residue in an oxidoreductase reaction-specific manner, as well as fragments that induce significant structural rearrangements of crucial regulatory elements. Our findings establish a framework for extending CFS beyond traditional inhibitor discovery, demonstrating its utility in probing protein dynamics, identifying novel binding pockets, and investigating regulatory mechanisms. These results offer new insights into DHS function, hypusination dynamics, and the broader methodological advancements that CFS contributes to structural biology and protein regulation research.

Hypusination is a unique post-translational modification of a lysine residue in eukaryotic translation factor 5A (eIF5A), crucial for resolving polyproline-induced ribosomal stalling, facilitating the clearance of colliding ribosomes[1], and contributing to the regulation of translation speed[2]. Notably, eIF5A functions in translation only when activated through hypusination at a specific lysine residue (K50 in *Homo sapiens*). Despite its rarity, this modification requires the concerted action of two enzymes: deoxyhypusine synthase (DHS)[3–6] and deoxyhypusine hydroxylase (DOHH)[7]. The first rate-limiting step of the reaction is catalyzed by NAD-bound DHS, which transfers a 4-aminobutyl moiety derived from spermidine (SPD) to the K50 of eIF5A via an intermediate state on K329 of DHS[3] (Supplementary Fig. 1). Deoxyhypusinated eIF5A is further hydroxylated

by iron-dependent DOHH to form the final, mature form of hypusinated eIF5A.

Hypusine was initially isolated from bovine brain extracts in 1971. A decade later, in 1981, the discovery of eIF5A established it as a hypusine-containing protein[8]. Subsequent studies progressively enhanced our understanding of eIF5A's role in cellular processes and human health. Notably, eIF5A remains the only protein known to contain hypusine[9]. Further research revealed that hypusine is formed post-translationally as a modification of a lysine residue and is not incorporated as a free amino acid. The study of its regulation is ongoing, though it's currently hindered by a limited experimental toolbox. Nonetheless, it is well documented that disruptions in the hypusination process, including impaired function of the

[1]Malopolska Centre of Biotechnology, Jagiellonian University, Kraków, Poland. [2]Jerzy Haber Institute of Catalysis and Surface Chemistry Polish Academy of Sciences, Kraków, Poland. [3]Department of Organic Chemistry, Faculty of Chemistry, Jagiellonian University, Kraków, Poland. [4]Doctoral School of Exact and Natural Sciences, Jagiellonian University, Kraków, Poland. [5]MAX IV Laboratory, Lund University, Lund, Sweden. ✉e-mail: przemyslaw.grudnik@uj.edu.pl

hypusination machinery involving DHS and DOHH, have been linked to the etiopathology of various diseases, such as certain cancers and neuro-developmental disorders[10]. In cancer, elevated levels of hypusinated eIF5A are observed, and inhibition of the hypusination pathway has been shown to suppress tumor growth and progression[11–16]. Similarly, in diabetes, hypusination plays a critical role in maintaining β-cell function, and disruptions in this pathway can lead to β-cell dysfunction and the onset of diabetes[17–20]. Additionally, increased hypusination has been linked to pulmonary arterial hypertension, where it contributes to the hyperproliferation of pulmonary arterial smooth muscle cells[21]. Furthermore, reduced eIF5A activity has been linked to several disorders, with strikingly similar clinical manifestations whether caused by mutations in the eIF5A encoding gene itself[22] or by defects in the hypusination pathway enzymes[23,24]. Although correlations between eIF5A hypusination and various diseases have been established, in-depth investigations remain challenging due to the lack of molecular probes and potent, highly specific inhibitors of this process[25–28].

A detailed molecular understanding of hypusination is crucial for rational drug design. Previously, we elucidated the structural basis of DHS substrate selectivity, particularly its ability to discriminate between SPD and other polyamines[6]. DHS functions as a tetramer, composed of two tight dimers, each containing two equivalent active sites (Fig. 1a). Access to each active site is possible only through a narrow tunnel, which entrance can be sealed by the ball-and-chain (BnC) motif—formed by the N-terminal region of the opposite dimer (Fig. 1b).

Our findings revealed that in the absence of its cofactor, NAD, the overall dimer-of-dimers architecture of DHS remains intact. However, in the holoenzyme—where all active sites are occupied by NAD molecules and SPD (Fig. 1c)—the highly flexible 30-residue N-terminal region becomes partially ordered and traceable in one subunit of the homodimer (Fig. 1a). This BnC motif binds over the entrance to the active site (Fig. 1b) in the opposite dimer, playing a crucial regulatory role. Its deletion not only abolishes DHS activity but also shifts its oligomeric state from a tetramer to a dimer[6].

Structural analysis of the eIF5A-DHS complex confirmed a 1:4 stoichiometry, demonstrating that eIF5A competes with the BnC motif for access to the active site[29] (Fig. 1d). More recently, DHS was shown to interact with Raf/MEK/extracellular regulated kinases 1/2 (ERK1/2), which also compete with eIF5A for binding[30] (Fig. 1e). Furthermore, the capture of a transition state analog has provided a more comprehensive in vitro understanding of the deoxyhypusination mechanism[29]. However, without selective inhibitors or molecular probes, it remains unclear how these in vitro findings translate to cellular and in vivo processes. Broad-spectrum inhibitors like N1-guanyl-1,7-diaminoheptane (GC7) affect multiple pathways beyond hypusination, making it difficult to isolate its specific role.

Initial attempts at DHS inhibition were performed in the early 90s[31–33], and additionally helped to establish the proposed binding model of SPD in the active site. Several bis- and mono-guanylated polyamines and their derivatives were investigated. The most potent compound identified was N1-guanyl-1,7-diaminoheptane (GC7), exhibiting a Ki value of approximately 10 nM. Due to its high efficacy, GC7 has since become the gold standard in studies of hypusination, despite being a SPD analog and consequently exhibiting a notable level of off-target activity against other polyamine-binding proteins. Importantly, further studies confirmed that GC7 is actively transported into the cell using the polyamine transport system. This was demonstrated by the fact that mutant CHO cells defective in polyamine transport were resistant to GC7's growth inhibition, confirming that the compound leverages the natural polyamine uptake mechanism to reach its intracellular target[32]. Additional studies on tumorigenic cell lines showed that GC7 acts as a general antiproliferative agent. This effect is supported by the hypothesis that the synthesis of proteins essential for S-phase progression is dependent upon the presence of hypusinated eIF5A[34]. In 2020, Tanaka and colleagues published two distinct optimized amide compounds derived from high-throughput screening (HTS). Based on their inhibition studies, they hypothesized that their bromobenzothiophene compound competitively occupies/rearranges the

binding site of both NAD and SPD. Considering that DHS is constantly bound to NAD/NADH in the cell, switching between the NAD-bound and NADH-bound states, this mechanism may not represent the optimal strategy for inhibition[28,35]. Recently, Liu and colleagues shared the series of oxadiazole derivatives with superior efficacy to GCY in A375 zebrafish model, but based on the docking experiments occupying again SPD binding sites[36]. Moreover, efforts continue to develop novel screening platforms that facilitate the search for new and more specific inhibitors[37–39]. All these challenges underscore the critical need for further investigation into the hypusination pathway and the development of related compounds—especially since a diminished DHS activity has been linked to rare genetic disorders[22,23], highlighting the potential therapeutic significance of both inhibitor and deoxyhypusination activator research. Fragment screening is a method used to identify small, typically weak-binding organic compounds that interact with a target protein. These identified binders serve as building blocks or "fragments" for designing follow-up ligands with increased affinity. Compared to conventional HTS libraries, fragment libraries are several orders of magnitude smaller but are designed to cover a similar extent of chemical space[40–44].

Various techniques with differing throughput and sensitivity have been employed in fragment screening campaigns[45], but one approach stands out: X-ray crystallographic analysis. Unlike other methods, crystallography not only provides a binary yes/no identification of fragment binding but also delivers detailed structural information on the fragment's position and orientation within the protein. Furthermore, when analyzed against hundreds of structures, this technique enables the identification of clusters of fragments binding in spatially adjacent regions, an invaluable advantage for designing second-generation binders[46,47].

Historically, macromolecular crystallography (MX) faced limitations in throughput and sensitivity, particularly for detecting weak binders. However, recent advancements in beamline automation, data handling, and analysis algorithms have significantly improved these aspects[48,49]. As a result, MX has emerged as a leading technique for fragment screening, offering unparalleled precision and insight into protein-ligand interactions.

In this study, we used crystallographic fragment screening (CFS) to explore novel small-molecule binders of human DHS. Initially performed on the apo form, the campaign was refined by including NAD, improving hit identification and structural clarity. We identified fragments stabilizing the regulatory BnC motif, targeting the active site entrance, and binding at the tetramer interface, revealing new peripheral, potentially allosteric, binding sites. Additionally, a covalent modifier of the DHS's catalytic lysine suggested a mechanism-based inhibition strategy. These findings provide insights into DHS regulation and highlight new avenues for therapeutic targeting of hypusination.

## Results
### Cofactor binding stabilizes DHS structure and enhances fragment screening success
To identify novel fragment binders of human DHS, we optimized crystallization conditions and tested the crystal stability in the presence of dimethyl sulfoxide (DMSO). Crystals were soaked with the FragMAXLib fragment library composed of 172 compounds (Supplementary Data 1), and duplicates for each fragment were harvested and flash-cooled in LN$_2$. Diffraction data were collected at the BioMAX crystallography beamline at the MAX IV synchrotron. Data processing and hit identification were performed using FragMAXapp[49] and PanDDA[48,50].

Initially, we conducted fragment screening on DHS in its apo, NAD-free form to ensure fragments could access the NAD-binding site. Unlike previously described Trichomonas vaginalis DHS, which co-purifies with NAD[51], human DHS was obtained in a high-purity NAD-free state. Unfortunately, the data interpretation was significantly less straightforward than expected for a seemingly homogenous set of structures at average 1.7 Å resolution. It appeared that almost all identified PanDDA events were false positives coming either from a partly ordered N-terminus (absent in the apo DHS search model) or predominantly from disordered loops forming an

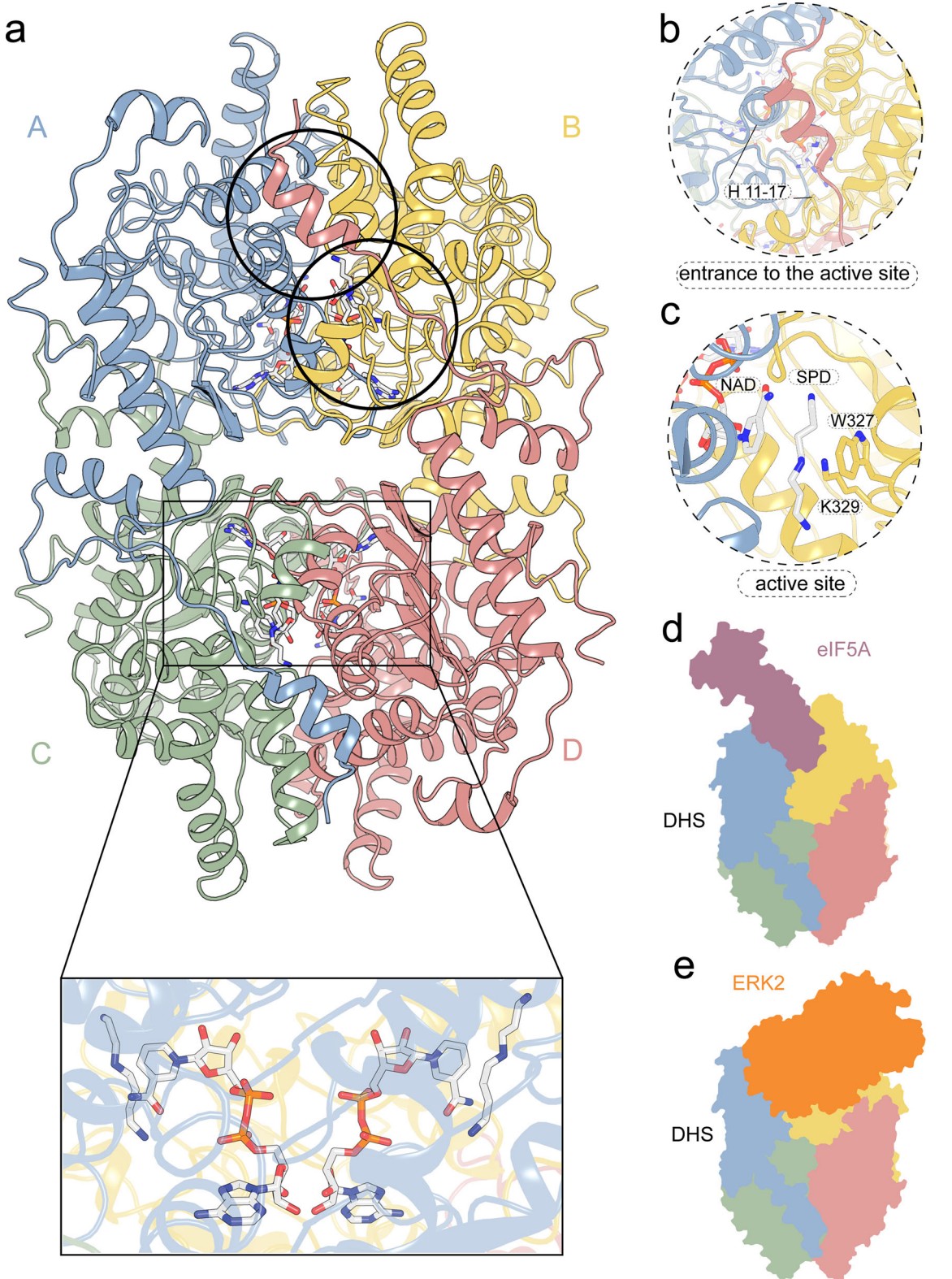

**Fig. 1 | Overall architecture of DHS. a** Cartoon representation of the homotetrameric human DHS, composed of two tightly associated dimers. Each protomer is colored individually. The inset shows the location of the active site, marked by an NAD cofactor and two spermidine molecules, related by a two-fold axis and buried deep within the protein core. **b** The entrance to the active site is occluded by an N-terminal ball-and-chain motif originating from the opposing dimer. **c** Key

components of the active site include the NAD cofactor, spermidine substrate, the catalytic lysine K329, and a mobile tryptophan residue (W327) that seals the reaction center from the bulk solvent. **d, e** Schematic representations of DHS complexes. The ball-and-chain seal can be displaced either by the eIF5A substrate bound in a 1:4 stoichiometry (**d**) or by the ERK2 kinase (**e**).

NAD-binding Rossman fold. (Supplementary Fig. 2 and Supplementary Movies 1 and 2).

To improve screening success and enhance data interpretability, we repeated the campaign under more stabilizing conditions by including NAD during protein preparation. Given that, NAD-mimicking inhibitors would likely be nonspecific due to the NAD's ubiquity as a cellular cofactor and its tight binding to DHS, we concluded that fragments targeting this site would have limited therapeutic applicability. However, the presence of NAD significantly improved structural resolution and interpretability by stabilizing previously disordered loops. This stabilization underscores the critical role of cofactors in fragment screening and highlights the importance of protein preparation in structural studies of enzymatic targets.

In our CFS campaign, we identified 67 unique fragments bound to DHS out of the 172 fragments used for soaking. The total number of binding sites was 136, reflecting the presence of a protein dimer in the asymmetric unit. Importantly, for many lower occupancy ligands, the main evidence for the binding event comes from the PanDDA[48] event maps, which are shown for all identified ligands in Supplementary Data 2. Notably, some regions of the monomers exhibited different levels of ordering; for example, the BnC motif was typically well-ordered in chain A but more flexible in chain B. In most cases, the same fragment was observed twice in the asymmetric unit, bound to the corresponding regions of both monomers. However, 17 fragments were identified in multiple distinct locations (VT00015; VT00025; VT00048; VT00049; VT00079; VT00155; VT00165; VT00178; VT00193; VT00215; VT00217; VT00228; VT00234; VT00259; VT00407; VT00445; and VT00451).

Overlaying the bound fragments onto the DHS structure revealed several distinct binding clusters (Fig. 2a, b): the active site, entrance tunnel, BnC motif, tetramer interface, and allosteric sites. Each of these clusters offers unique opportunities for functional exploitation in follow-up studies.

Among these clusters, the peripheral binding sites exhibited the highest number of unique fragments, with 27 distinct fragments across 37 binding sites (Fig. 2c). The BnC motif also contained 37 binding sites but fewer unique fragments (24). The tetramer interface displayed a significant number of symmetrical fragments, with 18 unique fragments at 33 binding sites. The entrance tunnel leading to the active site hosted 14 unique fragments across 21 binding sites. Finally, the fewest fragments were bound to the active site itself, with just 4 unique fragments occupying 8 binding sites.

### Fragment binding induces major conformational changes in the regulatory ball-and-chain motif

In eukaryotes, DHS forms a tetrameric complex with its substrate eIF5A in a 4:1 ratio (Fig. 1d), where three of the four active sites are sealed by the BnC motif[29]. This motif plays a crucial role in regulating the enzymatic activity and stoichiometry of the complex by modulating access to the active site[5,6]. Previously, we demonstrated that altering BnC affinity to the active site entrance directly affects DHS catalytic efficiency. For example, the naturally occurring pathological DHS[N173S] variant[23] exhibits significantly lower activity, probably due to a disrupted hydrogen bonding network between BnC and DHS core.

Our fragment screening campaign identified 24 unique fragments binding at 37 sites near the BnC motif (Fig. 2c). Many of these fragments appear to stabilize this motif in a conformation that effectively blocks access to the active site. While analyzing PanDDA-identified regions of interest, we observed several instances where the fitted ligands did not fully account for the electron density. Further analysis revealed that this density corresponded to a repositioned BnC motif, stabilized by bound fragments, allowing us to trace the DHS N-terminus in a previously unobserved orientation (Figs. 3 and S5).

Notably, several fragments induced extensive structural rearrangements of the BnC motif, stabilizing it in one DHS subunit and coupling this stabilization across both observable subunits (Fig. 3a). In the original orientation (known e.g., from previously published DHS crystal structure in complex with NAD and SPD) the N-terminal helix is closing the entrance to the active site in the opposite dimer (Fig. 3b) In the stabilized orientation, the

BnC motif rests along the dimer interface, leaving the active sites exposed to bulk solvent (Fig. 3c). This exposure could potentially facilitate a futile side reaction, leading to the cyclization of SPD into $\Delta^1$-pyrroline[9]. Interestingly, this conformation resembles one recently reported for archaeal *Sulfolobus islandicus* DHS, where the significantly shorter N-terminus assembles near the enzyme's tetrameric interface, leaving all active sites accessible for aIF5A binding[52,53].

These structural shifts indicate that small-molecule ligand interactions can influence key regulatory checkpoints of DHS activity. Beyond deepening our understanding of DHS function, these findings open new avenues for the development of protein activity modulators specifically targeting the BnC motif.

### Multiple fragments cluster around the entrance to the active site

The catalytic core of DHS, where the NAD cofactor and the catalytic K329 residue reside, is deeply buried within the enzyme and connected to the surface by a narrow tunnel[3]. This tunnel is specifically designed to accommodate the hypusine loop of eIF5A during catalysis[29]. In the absence of the protein substrate, the tunnel is sealed at the bottom by the sidechain of W327, while the entrance is regulated by the N-terminal BnC motif (Fig. 1a, b).

Fragment screening revealed a notable clustering of ligands at the entrance of this tunnel. We identified seven distinct fragments (VT00048, VT00165, VT00190, VT00234, VT00257, VT00447, and VT00451) binding in proximity to each other, forming a network of interactions that could potentially interfere with substrate access (Fig. 2c). As previously observed, human DHS crystallizes asymmetrically, with one BnC motif ordered and covering the entrance to the active site, while the second active site remains exposed to solvent. In this open conformation, an additional seven fragments (VT00025, VT00173, VT00204, VT00268, VT416, VT00423, and VT00445) were detected within the entrance tunnel, suggesting that this region is particularly prone to small molecule binding (Fig. 2).

The presence of multiple fragment hits in this critical region suggests a strong potential for modulating DHS activity by interfering with substrate binding. Fragments binding at the tunnel entrance could serve as a foundation for designing molecules that either stabilize an inactive conformation of DHS or selectively disrupt substrate processing. Given the essential role of this tunnel in catalysis, these findings offer a promising avenue for developing inhibitors that target the enzyme's substrate accessibility rather than competing directly with cofactor or active site interactions.

### Discovery of covalent catalytic modifiers

The active site seems to be the most attractive binding location for potential inhibitor precursor, competitive for binding with the substrate, SPD, occurs. Binding of SPD occurs deep in the active site in the close proximity of NAD molecule, catalytic K329 and W327, which as we showed previously has role in the transfer of 4-aminobutyl moiety during reaction[29] (Fig. 4a). One known inhibitor, molecule N1-guanyl-1,7-diamine-heptane (GC7), binds to the active site in the same manner as SPD (Fig. 4b); however, being a substrate analog also results in numerous off-target effects. The catalytic mechanism of DHS involves an initial electron transfer step, where NAD is reduced, followed by a transient covalent attachment of the 4-aminobutyl moiety of SPD to the catalytic lysine residue K329. This covalent intermediate is a crucial step in the hypusination process, facilitating subsequent transfer of the 4-amionbutyl to the acceptor K50 at the incoming eIF5A, leading to formation of deoxyhypusine. Next, this product dissociates from the active site and can undergo a subsequent hydroxylation by DOHH. In our previous studies, artificial stabilization of this intermediate enabled us to capture a stable transition state analog, where K329 remained deoxyhypusinated (PDB ID 8A0G)[29]. Building upon this knowledge, we sought to determine whether fragments identified in our screening campaign could directly interact with or modify this key catalytic residue.

While inspecting PanDDA maps from fragment-soaked DHS crystals, we identified a strong peak in the event map for the crystal structures hsDHS-x0037 and hsDHS-x0038, both soaked with fragment VT00065.

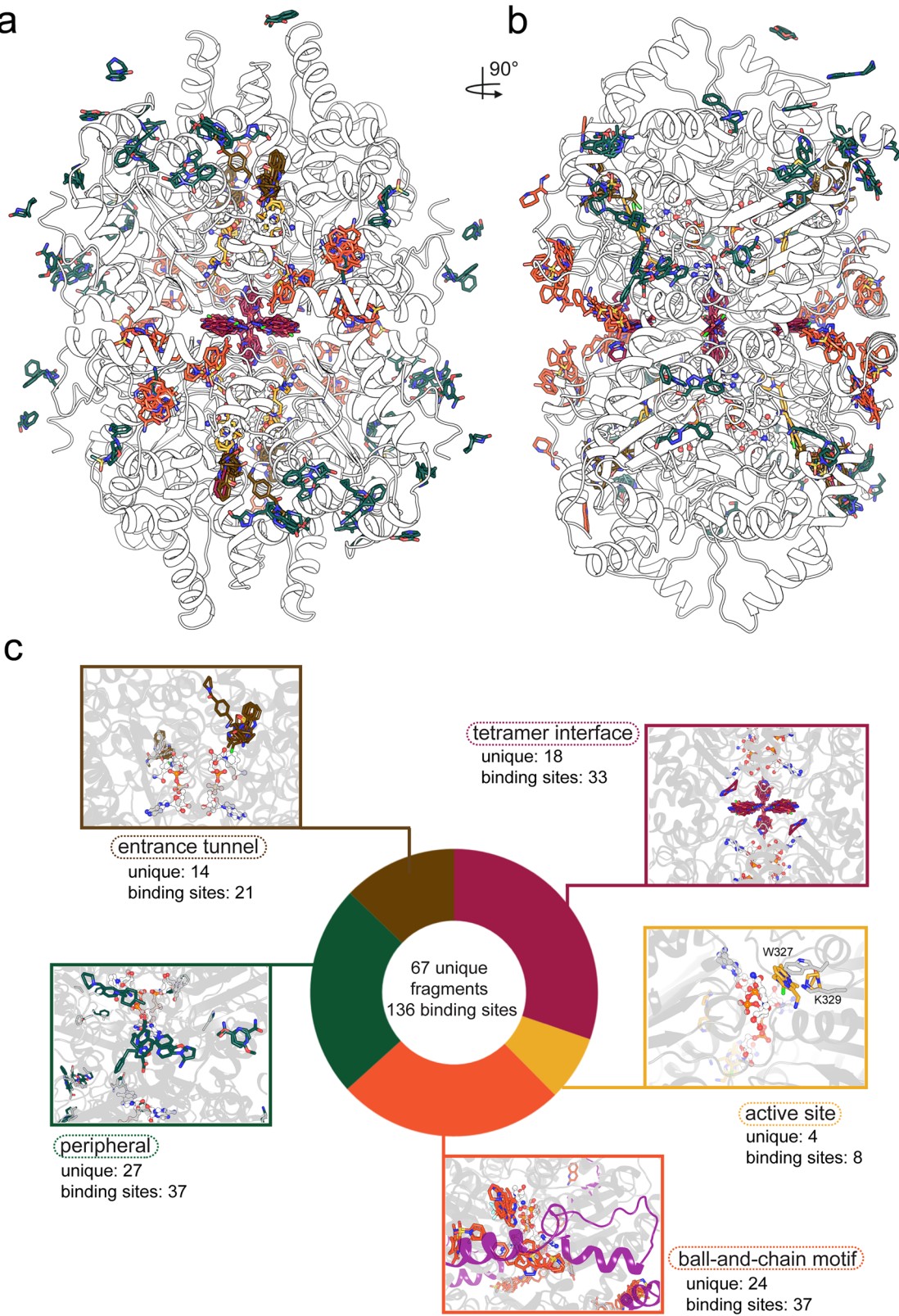

**Fig. 2 | Summary of the fragment screening campaign. a, b** Visualization of the ensemble of identified ligands mapped onto the DHS structure, shown in two perpendicular views as a white cartoon. Ligands are shown as sticks, with carbon atoms colored according to ligand classification. **c** Classification of identified hits, with representative ligand clusters projected onto the structure. The coloring scheme for ligand carbon atoms is as follows: active site–yellow; entrance tunnel– brown; ball-and-chain motif–orange; interface–purple; peripheral sites–green. This color convention is maintained throughout the manuscript. The number of hits within each classification is indicated for each cluster. Ligands were counted per DHS dimer present in the crystal asymmetric unit (ASU); some ligands bind in more than one pocket, so the total number of binding sites exceeds the number of unique ligands.

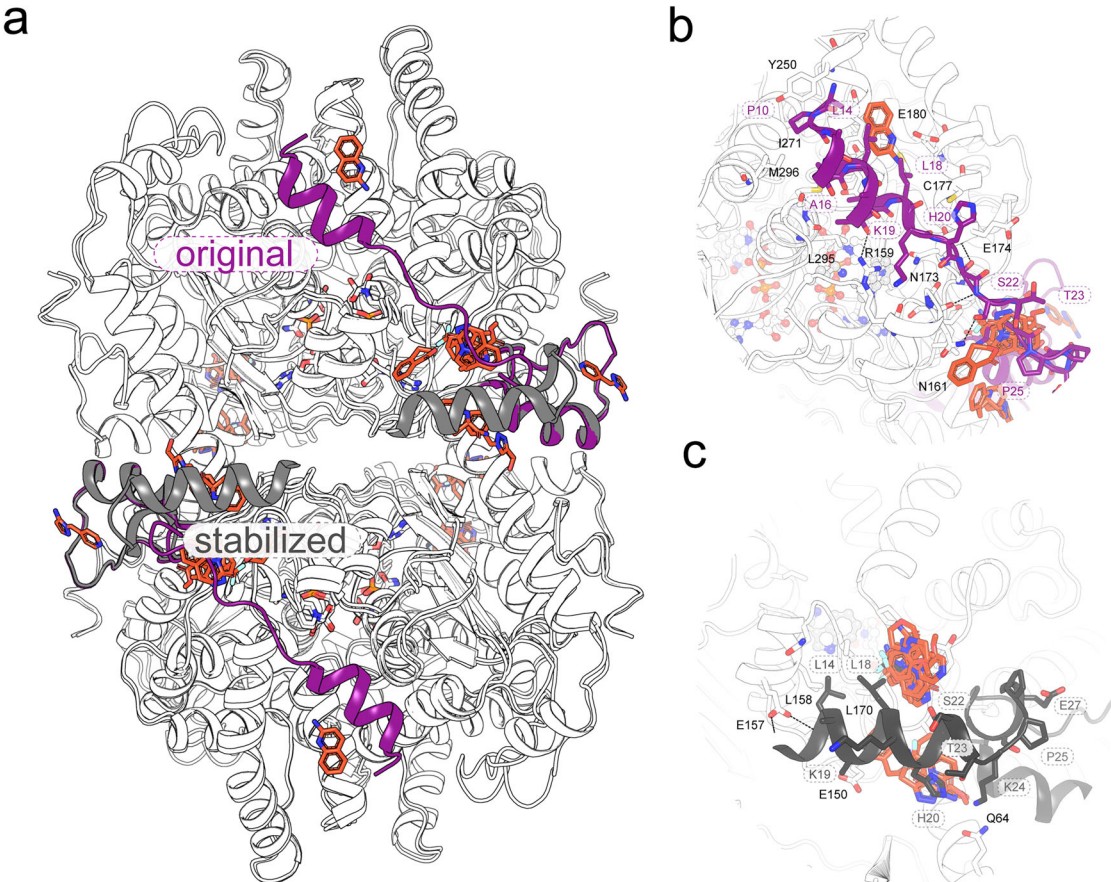

**Fig. 3 | Induced rearrangement of the ball-and-chain motif.** The N-terminal peptide of DHS is highly flexible. In the presence of NAD and spermidine (SPD), it becomes stabilized, locking half of the active sites. In complex with eIF5A, it locks all active sites except the one occupied by the target protein. **a** Superposition of DHS structures showing the ball-and-chain motif in its physiological orientation over the active site entrance (magenta) and in a rearranged pose along the tetramer interface (gray). **b** Canonical interaction of the ball-and-chain with the DHS core. Interacting residues are labeled for the ball-and-chain (magenta) and the core protein (black). **c** Interaction of the rearranged ball-and-chain with the DHS core. Interacting residues are labeled for the ball-and-chain (gray) and the core protein (black). Ligand clusters bound near each orientation are shown as orange sticks.

The density was located directly adjacent to the K329 Nε amine, raising the possibility of a covalent interaction. Attempts to model the ligand within this density revealed a well-defined thiazole ring orientation; however, the aliphatic portion of VT00065 collided with the lysine side chain in such a way that their nitrogen atoms were nearly perfectly aligned (Fig. 4c). Notably, no similar event density was observed around any other lysine residue in the structure, suggesting a mechanism-based, redox-dependent cleavage of the fragment with the simultaneous formation of a covalent bond with K329 (Fig. 4d).

This finding strongly suggests that VT00065 functions as a covalent binder, irreversibly modifying the catalytic lysine and potentially locking DHS in an inactive state. However, the absence of clear electron density in the 2fo-fc map indicated that the occupancy was low, suggesting weak binding affinity. Despite this, the specificity of the modification to K329 highlights the potential of mechanism-based inhibitors for DHS, offering an attractive strategy for highly selective drug development.

In line with the crystallographic observation are, in vitro analyses of the rate of hypusination in the presence of VT00065 assessed by Western blot using a deoxyhypusine-specific antibody. In this assay, VT00065 shows some signs of inhibitory properties (Fig. 4e).

Beyond VT00065, three additional fragments (VT00015, VT00154, and VT00409) were identified in the active site (Fig. 2c), although their interactions were less clearly resolved. These findings provide a valuable foundation for the future development of covalent inhibitors targeting DHS, with potential applications in both fundamental research on hypusination

and therapeutic intervention. Further optimization of these fragments, particularly to enhance binding affinity and reactivity, could yield a new class of irreversible DHS inhibitors with high specificity and therapeutic potential.

### Targeting the tetramer interface and design of the follow-up compounds

DHS functions as a tetramer, forming a dimer of dimers, a structural arrangement essential for its enzymatic activity. In our previous studies[6], we demonstrated that disruption of this oligomeric state leads to enzyme instability and a loss of catalytic function. Given the crucial role of the tetramer interface in DHS activity, we hypothesized that this region could serve as a promising target for small-molecule modulators. Fragment screening revealed that this extended interface provides an attractive binding site for small molecules, with multiple fragments identified at distinct regions within the oligomeric assembly.

Two major fragment-binding clusters were observed within the tetramer interface. The first cluster, located at the core of the DHS tetramer, was positioned between the ribose moiety of NAD and residues T308, E310, F312, A341, and their symmetry mates. Fragments bound in this region (VT00049, VT00082, VT00089, VT00222, VT00224, VT00228, VT00236, VT00403, VT00427, VT00423, VT00424, and VT00438) were observed in a four-fold symmetry, forming a cross-like ensemble, suggesting a strong preference for this site (Fig. 2a, b). The second cluster was located closer to the protein surface, near the crystallographic two-fold axis, involving

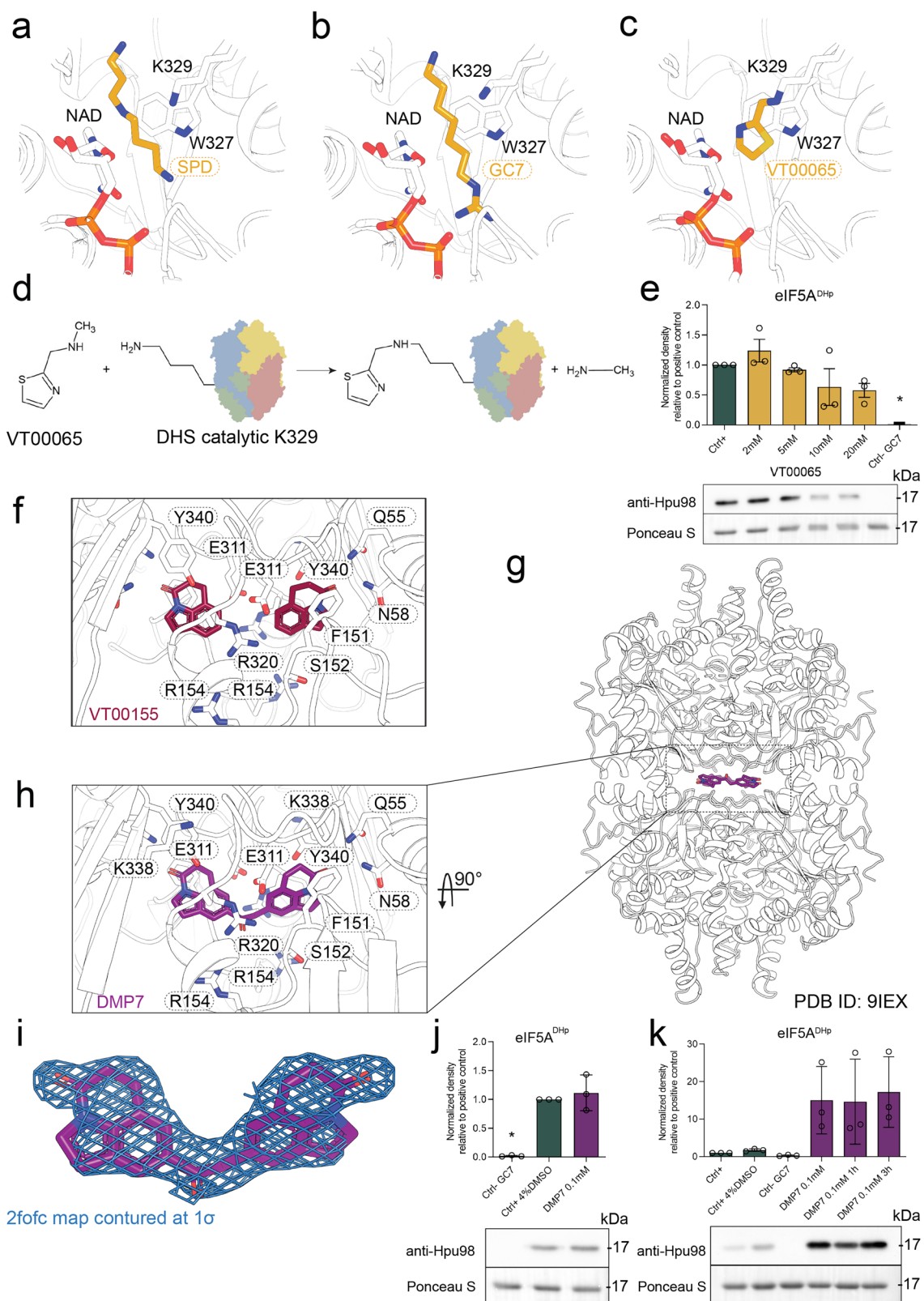

PDB ID: 9IEX

2fofc map contured at 1σ

residues N58, E150, F151, S152, E311, R320, and Y340 of both monomers. Interestingly, ligands bound in this region (VT00048, or VT00155, VT00175, VT00217, VT00218, VT00224, VT00230, VT00236, VT00254, VT00427, and VT00438) were in close proximity to both the four-fold symmetry cluster and their symmetry-related counterparts, making this site particularly attractive for fragment linking and ligand optimization.

Notably, the tetramer interface displayed the highest number of clustered hits identified in our screening campaign, reinforcing its potential as a target for modulating DHS activity (Fig. 2c).

We decided to explore the potential of fragment-based compound design by linking two molecules present in the close proximity to one another. Many such instances were observed in the vicinity of the interface

**Fig. 4 | Specific targeting of DHS.** Active site with bound **a** spermidine (SPD) **b** N1-guanyl-1,7-diaminoheptane (GC7), and **c** VT00065 covalently linked to K329. **d** Proposed simplified mechanism of reaction leading to linking of VT00065 to the side chain of lysine. **e** In vitro deoxyhypusination assay with Western blot readout demonstrating weak but dose-dependent inhibitory effect of VT00065. **f** VT00155 fragment (pyroquilone), shown as sticks, is bound to the DHS interface with two molecules in close proximity, suggesting the possibility for linking. **g** DMP7 molecule bound at the DHS tetramer interface is shown as purple sticks. **h** The close-up view on the DMP7 and its binding pocket with residues involved in binding shown as sticks. **i** Electron density map shown as blue mesh contoured at 1σ level. **j** Western blot-based DHS activity assay showing no impact of DMP7 on the catalytic activity of DHS. **k** The same assay as in (**j**), showing stabilizing effect of DMP7 after prolonged incubation of DHS in the assay conditions (control samples were also incubates at RT for 3 h before analysis). For **e**, **j**, **k**: for deoxyhypusination detection, monoclonal rabbit FabHpu98 (#PABL-582) antibody was used (Creative Biolabs). Data are mean ± SEM of biological triplicates where "*" indicates a statistically difference from the control sample with $P < 0.005$ determined by one-way ANOVA with post hoc Dunnett's tests. Below graphs, representative Western blots and loading controls are shown. All of the replicates with statistical analysis are presented in Supplementary Figs. 3 and 4.

with pyroquilone (VT00155, Fig. 4h), standing out as one of the strongest hits, visible in both PanDDA and conventional maps. The resulting compound, DMP7, was engineered to span two DHS dimers normally related by a crystallographic two-fold axis (Fig. 4f–i). The structural analysis confirmed that DMP7 has bound in the expected location (Fig. 4i), however due to the asymmetric nature of the introduced linker its binding resulted in a reduction of crystal symmetry from $P3_221$ to $P32$ (see Supplementary Data 3). Two copies of DMP7 are present in a HsDHS crystals structure (Fig. 4f–i). The ligands are located symmetrically at the interface, approximately 14 Å from the center of the tetramer. Each copy of the DMP7 ligand interacts with all DHS subunits, thereby making crucial aliphatic-pi stacking interactions between its pyroquilone moieties and F151 (c. A) and Y340 (c. D) on one side and F151 (c. C) and Y340 (c. B). The binding site is completed by symmetrically arranged sidechains of hydrophilic residues (E55, N58, and E311) and the main chain of S152 and F312. Importantly, the carbonyl (O1) from the linker is sandwiched between guanidinium moieties of R154 and R320, what may be an important selectivity determinant. Despite its successful binding, in vitro assays showed that DMP7 did not exhibit significant inhibitory effects on DHS activity (Fig. 4j). On the contrary, the western blot-based analysis shows a higher level of deoxyhypusination in the reaction when preincubated with DMP7 (Fig. 4k). This effect may be attributed to the stabilization of DHS by the ligand.

Nevertheless, targeting the tetramer interface remains an unexplored avenue for enzyme modulation. The ability to stabilize or destabilize the oligomeric state of DHS presents an opportunity to regulate its function without directly competing with substrate or cofactor binding. This strategy could be particularly valuable for designing selective inhibitors with improved specificity. Moreover, molecules targeting this interface could serve as warheads for proximity-inducing drugs, such as PROTACs or molecular glues, expanding their potential applications in therapeutic development. Further optimization of these compounds, particularly through rational ligand expansion and structure-guided drug design, may facilitate the development of novel DHS modulators with enhanced potency and specificity.

### Surface-binding fragments enable mapping new druggable pockets

While the primary function of DHS is the post-translational modification of eIF5A, recent studies have demonstrated its tight interaction with ERK1/2[30], suggesting that DHS may have additional, yet undescribed, regulatory roles beyond hypusination (Fig. 1e). Understanding these potential interactions requires tools that selectively probe protein–protein interactions (PPIs) and modulate DHS function in a controlled manner. Our fragment screening campaign revealed several unique ligands binding to the surface of DHS, providing potential scaffolds for such tools.

Unlike fragments identified at the active site or tetramer interface, surface-bound fragments displayed more variable binding patterns. Some ligands were symmetrically present in both observed chains, while others were detected in only one chain, potentially due to crystal packing effects. Many of these surface-bound fragments were found close to each other, forming clusters that suggest additional, previously unrecognized binding pockets. Although these fragments may not be directly applicable for developing catalytic inhibitors of DHS, they hold significant potential for designing molecular probes targeting regulatory interactions (Fig. 2).

One of the promising applications of surface-binding fragments is in the development of proximity-labeling probes, such as proteolysis targeting chimeras (PROTACs)[54] or molecular glues[55,56], which can facilitate targeted protein degradation or stabilization. The ability of these fragments to interact with surface-exposed motifs suggests that DHS could be selectively modulated by small molecules designed to interfere with or enhance its interactions with other cellular partners. Given the emerging role of DHS in broader signaling networks, particularly concerning ERK1/2, these fragments provide valuable starting points for mapping DHS interaction sites and identifying novel druggable pockets.

### Discussion

Our fragment screening campaign provides a comprehensive structural map of DHS, identifying multiple binding sites that offer new opportunities for mechanistic studies and drug development. The diverse range of discovered fragment interactions highlights the complexity of DHS regulation and suggests several distinct approaches for modulating its function.

One of the intriguing aspects of DHS regulation is its oligomeric state. All known DHS enzymes function as a dimer of dimers, a conserved arrangement observed across diverse species, including *Sulfolobus islandicus*[57], *Trichomonas vaginalis*[51], and humans. In contrast, in some pathogenic protozoa such as *Trypanosoma brucei* and *Leishmania donovani*, DHS forms heterodimers containing an inactive paralogue[58]. This suggests that oligomerization plays a crucial role in enzyme regulation, and in some species, an inactive paralogue may act as a built-in regulatory subunit. Our fragment screening identified multiple binding sites at the DHS tetramer interface, supporting the idea that perturbing this interface could influence enzymatic activity. Given our previous findings that dimeric DHS is unstable and inactive, targeting the tetramerization interface presents a novel approach for modulating enzyme function. In human disorders caused by a loss-of-function mutation, even a partial restoration of enzymatic activity could be highly beneficial for affected patients[59,60]. The DMP7 compound described here exhibits a beneficial effect on enzyme performance, likely through stabilization, and presents a promising avenue for investigating DHS-deficiency-related variants[23,61,62]. Future efforts could focus on designing small molecules that either stabilize or disrupt this interface, offering an alternative approach to enzyme inhibition without competing for substrate or cofactor binding. Notably, the identified ligands could also be developed into specific fluorescent probes, enabling antibody-independent quantification and subcellular localization of DHS.

The most promising approach for developing enzyme-specific inhibitors is to target its active site (AS) or the tunnel connecting it to the bulk solvent. In DHS, the active site is located deep within the protein core and is accessed by the substrate peptide through a relatively deep and narrow entrance tunnel. The W327 side chain was previously shown to modulate substrate and product traffic to and from the AS.

We identified four fragments within the active site and an additional 15 unique fragments clustered in the entrance region (i.e., above the W327 side chain). Particularly interesting is VT00065 (N-methyl-1-(1,3-thiazol-2-yl) methanamine), which forms a covalent linkage exclusively with the side

chain of the catalytic K329 residue. We speculate that this occurs due to a redox reaction analogous to the first step of the natural DHS catalytic cycle, which involves the dehydrogenation of SPD with concomitant reduction of the NAD cofactor. Usually, the nucleophilic attack of K329 on deprotonated SPD results in the formation of an imine-enzyme intermediate, releasing diaminopropane. In VT00065, a similar imine-enzyme adduct is formed, with methylamine as the leaving group (Fig. 4d).

Since the reaction lacks a suitable acceptor, the second catalytic step cannot proceed, leading to irreversible enzyme inhibition. The fragment was identified in PanDDA event maps but is poorly visible in standard fo-fc and 2fo-fc electron density maps, suggesting low occupancy. Modifications aimed at increasing affinity, such as merging VT00065 with GC7, could enhance AS pocket saturation, improving the yield of covalent DHS inactivation.

Furthermore, our study expands the scope of DHS inhibition beyond traditional active site targeting. In drug discovery, competitive inhibitors often suffer from limitations, such as poor specificity due to the highly conserved nature of catalytic pockets. In contrast, our findings suggest that DHS could be targeted indirectly by disrupting its interactions with regulatory proteins. The identification of surface-binding fragments reveals previously uncharted regions of DHS that could be exploited for chemical biology applications. Instead of directly inhibiting DHS catalytic activity, small molecules could be designed to interfere with PPIs, modulate DHS stability, or alter its subcellular localization. Such an approach could enhance selectivity and specificity while reducing potential off-target effects commonly associated with active site inhibitors.

These findings open new avenues for developing molecular probes, such as PROTACs or molecular glues, that facilitate targeted protein degradation or stabilization. Furthermore, DHS's emerging role in broader signaling pathways, particularly its interaction with ERK1/2, suggests that modulating its activity could have implications beyond hypusination. The ability to selectively target DHS interactions may provide insights into its unexplored regulatory roles and offer new therapeutic strategies for diseases linked to dysregulated hypusination.

Overall, our study highlights the effectiveness of fragment-based approaches in uncovering novel druggable sites and regulatory mechanisms within DHS. By identifying fragments that bind not only to the active site but also to potentially allosteric regions, the tetramer interface, and previously unrecognized surface pockets, we provide a structural framework for future drug development efforts. These findings broaden the perspective on DHS as a therapeutic target, emphasizing the importance of structural dynamics, PPIs, and enzyme oligomerization in inhibitor design.

## Methods

### Protein expression and purification
The DHS protein was expressed and purified as previously described[6,29]. Briefly, the gene encoding full-length DHS (Uniprot: P49366, residues 1–369) was codon-optimized for expression in *Escherichia coli* (*E. coli*) and synthesized by Genescript. This gene was then cloned into the pET-24d plasmid using NcoI and BamHI restriction sites, incorporating an N-terminal 6×His-tag followed by a TEV protease cleavage site. The pET-24d vector carrying the desired sequence was introduced into *E. coli* Rosetta cells for expression. A single colony from an LB plate was used to inoculate an overnight culture, which was subsequently diluted 1:100 into terrific broth supplemented with kanamycin (50 µg/L) and chloramphenicol (35 µg/L). Bacterial growth was monitored until the culture reached an $OD_{600}$ of 1–1.2, at which point the temperature was reduced from 37 to 18 °C. Protein expression was induced by adding IPTG to a final concentration of 0.5 mM, and incubation proceeded overnight at 210 rpm. Cells were harvested via centrifugation ($17,700 \times g$, 12 min, 4 °C) and resuspended in lysis buffer (50 mM Tris-HCl, pH 7.8, 300 mM NaCl, 20 mM imidazole, 10% glycerol, 10 mM β-ME). Lysis was facilitated by sonication (15 min, 5 s pulse/3 s pause cycles) in the presence of lysozyme (Sigma-Aldrich), and benzonase (Sigma-Aldrich) was added to degrade nucleic acids. The lysate was clarified by ultracentrifugation ($53,000 \times g$, 45 min,

4 °C). The resulting supernatant was loaded onto a pre-equilibrated 5 mL HisTrap HP affinity column (GE Healthcare Europe GmbH, Freiburg, Germany). Unbound proteins were washed away using a buffer containing 50 mM Tris-HCl, pH 7.8, 200 mM NaCl, 40 mM imidazole, 5% glycerol, and 5 mM β-ME. The target protein was eluted with a high-imidazole buffer (50 mM Tris-HCl, pH 7.8, 200 mM NaCl, 400 mM imidazole, 5% glycerol, 5 mM β-ME). The purified protein underwent dialysis against a storage buffer (50 mM Tris-HCl, pH 7.8, 200 mM NaCl, 5 mM β-ME). During the second dialysis step, TEV protease was added to cleave the His-tag. The cleaved protein was subsequently purified using reverse HisTrap chromatography to remove any uncleaved protein and His-tagged TEV protease. Fractions containing the tag-free protein were concentrated using an Amicon Ultra concentrator (Millipore) with a 30,000 kDa molecular weight cut-off. The sample was further purified by size-exclusion chromatography on a HiLoad 16/60 Superdex 75 column in the storage buffer. The fractions exhibiting the highest purity were pooled, concentrated, aliquoted, and flash-frozen in liquid nitrogen for further use.

### Protein crystallization
DHS was concentrated to 20 mg/mL and supplemented with 5 mM NAD solution. The final protein solution was mixed in a 1:1 ratio with mother liquor based on Morpheus[63] screen (0.1 M Tris Bicine pH = 8.5, 47.5% precipitant mix 3 (mix of 25% of MPD, PEG 1000 and PEG 3350), 0.17 M carboxylic acid mix). DMSO-based fragment library was diluted 10x in the mother liquor, resulting in 100 mM fragments and 10% residual DMSO in the mother soaking drop. Crystals were incubated for 24 h with an excess of the fragment soaking solution in 10% DMSO. Soaked crystals were flash-cooled in $LN_2$ without further cryo-protection.

### Data analysis
The fragment screening campaign followed a standardized workflow, including crystal preparation, shifter-assisted soaking with the provided fragment libraries, harvesting, and flash-freezing in duplicate (excluding the apo form in this iteration). Data collection was performed at the BioMAX beamline, followed by automated data analysis using FragMAXapp[49,64]. Given that FragMAXapp generates multiple combinations of data processing, phasing, and analysis, leading to a potentially extensive set of models that differ only in minor details, a streamlined approach was adopted. Specifically, data processing and analysis were conducted using the following pipeline: XDSAPP–Dimple–ELBOW–PanDDA[48,50]. The initial quality assessment was performed by visualizing PanDDA event maps using UglyMol, integrated within FragMAXapp. Additionally, individual datasets were systematically reviewed using the panda.inspect script, which was adapted for Windows users. The detected blobs in event maps were cataloged, and identified events were subjectively scored based on their confidence level. Ligands with considerable confidence were modeled into the structure with partial occupancy, and the protein chain was adjusted as needed. The finalized models were subsequently refined using a standardized protocol in Refmac5. The processed data and unsorted PanDDA outputs were deposited in Zenodo for data reproducibility. A more detailed description of the protocol and a complete list of hits is provided in the Supplementary Methods and Supplementary Data File 2.

### Analysis of DHS catalytic activity
The in vitro deoxyhypusination assay was performed as previously described[30] with some modifications. Briefly, the reaction was done in 0.2 M glycine/NaOH pH 9.2 buffer supplemented with 1 mM DTT, 1 mM NAD, and 1 mM SPD at RT. 50 nM DHS was used, and the total reaction volume was 25 µl. DMP-7 stock solution was prepared in DMSO, so 4% DMSO was included in the appropriate control sample. To analyze the influence of the incubation time of DHS with a given compound on its effect on the activity of DHS, samples without SPD were incubated for 3 or 1 h at RT, and appropriate control sample was incubated for 3 h at RT. After that, SPD was added. The reaction was started by the addition of eIF5A1 (to a final concentration of 10 µM) with a multichannel pipette and after 9 min of

incubation stopped in the same manner with the addition of 5 μL o 6× SDS-page sample buffer. Samples were denatured at 95 °C for 5 min, and 6 μL of each sample was separated on 12% SDS-PAGE gel for Western blot analysis. Anti-(deoxy)hypusine (Hpu98) (Creative Biolabs #PABL-582) antibody[65] was used for immunodetection. Ponceau S signal was used for intra-assay chemiluminescent signal normalization. Densitometry was done with ImageJ[66]. Experiments were done in triplicate. Data are mean ± SEM, where "*" indicates a statistically significant difference from the control sample with $P < 0.05$, determined by one-way ANOVA with post hoc Dunnett's tests.

## Data availability

The 67 structures with bound fragments were deposited in the PDB with accession codes 9ID1 through 9IEV, and the structure with the follow-up compound DMP7 under the accession code 9IEX. Summary of data reduction and structure refinement statistics are provided in Table 1—like form as a Supplementary Data 3. The complete pool of automatically refined structures used for hit identification and PanDDA run parameters and outputs is provided as and Zenodo archive (https://doi.org/10.5281/zenodo.15059985). The fragment library may evolve slightly over time, and its composition as used in the experiment is provided as Supplementary Data 1. For the sake of potential design, a PyMOL session with all identified hits colored according to its classification is provided as Supplementary Data 4. Visualization of each hit with all its binding poses, along with their surrounding maps (ground state, standard and PanDDA maps) are provided as Supplementary Data 2. The coloring scheme employed for the fragments throughout the manuscript is yellow-gold for the active site, brown for the entrance to the active site, orange for the ball-and-chain related hits, magenta/purple for the interface located fragments, and dark green for surface/peripheral hits. The white ball-and-stick was used to depict native DHS residues and cofactors. Only carbon atoms were colored, and N, O, S, P, and Cl retained default colors, i.e., blue, red, yellow, orange, and green, respectively. Uncropped and unprocessed scans of the gels and Western blots are provided in the Supplementary Files.

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

## Acknowledgements

The research has been supported by the National Science Centre (NCN, Poland) research grant no. UMO-2019/33/B/NZ1/01839 and UMO-2022/47/B/NZ7/01667 to P.G. This research was partially funded by the project

"Inducing Molecular Proximity to Modulate Cellular Polyamine Metabolism" under the First Team FENG program of the Foundation for Polish Science (FNP), grant no. FENG.02.02-IP.05-0228/23. The project is supported by Priority 2 of the European Funds for a Modern Economy 2021–2027 (FENG). X-ray data were collected at the BioMAX beamline at MAX IV (Lund, Sweden) and at the XALOC beamline at ALBA Synchrotron Light Facility. We acknowledge the MAX IV Laboratory for beamtime on the BioMAX beamline under proposal 20200054 and ALBA synchrotron for the beamtime on the XALOC beamline under proposal 2023097899. Research conducted at MAX IV, a Swedish national user facility, is supported by Vetenskapsrådet (Swedish Research Council, VR) under contract 2018-07152, Vinnova (Swedish Governmental Agency for Innovation Systems) under contract 2018-04969 and Formas under contract 2019-02496. We would particularly like to acknowledge the support of Dr. Vladimir Talibov during our initial NAD-free fragment screening campaign during the COVID-19 pandemic. In addition, we thank the MCB Structural Biology Core Facility (supported by the TEAM TECH CORE FACILITY/2017-4/6 grant from Foundation for Polish Science) for providing instruments and support, in particular, Klaudia Woś, for her efforts and assistance during crystallization trials. E.W-W. and P.K. are supported by FNP START scholarships 2024 (E.W-W.) and 2025 (P.K.).

## Author contributions

P.G. initiated, conceived, and supervised the study. P.W. and P.G. designed the experiments. E.W-W. performed protein expression and purification. E.W-W., P.W., and T.K. crystallized protein, performed crystal soaking and fishing, and performed diffraction data collection; P.W. performed structure determination, refinement and analysis; P.W. and D.M. designed follow-up compounds; D.M. performed organic synthesis; P.K. performed Western blot analysis; P.W., E.W-W., P.K., T.K. and P.G. analyzed the data. E.W-W. and P.W. prepared figures. P.W. prepared the movies. P.W. and P.G. wrote the manuscript with assistance from the other authors.

## Competing interests

P.W., E.W-W., D.M., and P.G. are co-inventors of the Polish patent applications no. P.451617 and P.451618 claiming compounds described in this study. There are no other competing interests to declare.
