## [Transparent Peer Review File · Communications Chemistry]

Crystallographic fragment screening supports tool compound discovery and reveals conformational flexibility in human deoxyhypusine synthase

Corresponding Author: Dr Przemysław Grudnik

Version 1:

Reviewer comments:

Reviewer #1

(Remarks to the Author)

The authors have addressed the referees comments as outlined in the rebuttal letter. However while they may disagree with this referees comments, my opinion on this manuscript has not changed. The work reported is too preliminary to warrant publication in this journal. Further prioritisation of the fragment hits along with a biochemical or biophysical assay is needed as a minimum to progress this work to a stage that would be suitable for publication.

Reviewer #2

(Remarks to the Author)

The manuscript submitted to Communications Chemistry is a revised manuscript and contains the authors' rebuttals to the earlier reviewers' comments. The manuscript describes the results of screening human deoxyhypusine synthase against a library of 172 compounds using a fragment screening approach to identify the locations of potential inhibitors that might be lead compounds with future therapeutic value. The intended audience is for researchers in all areas of chemical sciences.

In response to the original comments, a rewriting of the introduction is appreciated and improves the readability of the manuscript, especially as it is being targeted to a community who are potentially less knowledgeable about biochemistry. Reviewer 1 suggested a mechanistic scheme, and given the intended audience, I think this would be a valuable addition, especially since there is reported to be a covalent inhibitor. The rebuttal addresses the concerns about the change in conformation of the N-terminal helix - the inclusion of the Fo-Fc electron density map for the N-terminus in supplementary figure 10 is noted.

The development of the DPM7 molecule is one of the key findings. I looked carefully at the data collection statistics and structure quality report as the electron density looked a little less well ordered than I might have expected. I noticed that it diffracts to lower resolution than the other datasets and that it is merohedrally twinned. The overall R_{merge} is 21% and R_{meas} is 22%, suggesting that the twinning may not have been handled in refinement. The experimental methods discuss the reason for the lower symmetry space group, but do not specifically state that the twinning was taken into consideration. I apologize for not noticing this in the initial review. However, the authors should address this either in the data analysis section of materials and methods, explaining why the values are so high if they did account for it, or re-refine the structure appropriately if they didn't.

In response to the rebuttal addressing PanDDA. I understand what the process is intended to resolve and am aware of the excellent work of Frank von Delft and his group. I would refer the authors to Pearce et al *Acta Cryst D Struct. Biol* (2017) vol73 pg 256-266, Gahbauer et al, *PNAS* v120 e2212931120 and the recent paper in *Nature Communications* (2025) v16 p8930. In the *Acta Cryst* and *Nat. Comms* papers, von Delft and his team provide great examples of fragment screening and the use of PanDDA to screen a 1076 fragment library. The hits they identify are more in line with the 20% or so typical hit rate. The PanDDA event maps in the papers show credible density for their compounds. Further, inspection of the structure reports for the compounds they identified show clear and appropriate 2Fo-dFc electron density maps and little, if any, mFo-DFc density contoured at 3 σ . This indicates that the molecules are present. By comparison, the same cannot be seen in event maps included in supplemental figure 5 for several of the compounds, for example – VT00219, VT00221, VT00228,

VT00229, VT00257, VT00405, VT00428.

All reviewers have raised concerns about the number of hits identified, characterization of VT00065 as a covalent inhibitor and the realism of the interactions. The authors rebut the reviewers' comments stating that their work is a proof of concept. In addition, they disclose that they are co-inventors of the compound, making publication important. Providing the question about twinning is resolved, there is sufficient supplemental data for an informed reader who is familiar with fragment screening to evaluate the data for themselves.

Reviewer #3

(Remarks to the Author)

The manuscript titled "Crystallographic fragment screening supports tool compound discovery and reveals conformational flexibility in human deoxyhypusine synthase" by Piotr Wilk et al. was previously submitted to Nature Com. As a reviewer I found this study particularly interesting. I had a few questions and comments, most of which have been addressed by the authors in their rebuttal letter.

The manuscript presents an engaging narrative on crystallographic fragment screening used to identify new inhibitors of DHPS. It also suggests potential new regulatory mechanisms by providing insight into the modification of DHPS conformation and it uncovers binding pockets. This work constitutes a proof-of-concept for a new method aimed at discovering hypusination inhibitors which, after further optimization, could potentially be used for treating various pathologies associated with eIF5A dysregulation such as cancer.

However, there is still one important issue that needs to be addressed.

In Fig 4e the authors performed an in vitro hypusination assay with increasing doses of VT0065. While the Western blot presented is convincing, the error bars between control condition and the highest inhibitor concentration overlap. This is confirmed by the p values provided in this new version of the manuscript (fig S9). Indeed, the p value between the ctrl and VT0065 is 0.25 which is usually not considered statistically significant (the commonly accepted threshold being < 0.05).

Thus, although the authors state that: "In this assay, VT00065 exhibits low millimolar inhibitory properties" the statistical analysis does not support the fact that an inhibitory activity, even low, is demonstrated by these experiments.

To resolve this, the authors should either increase the number of experiments to reach statistical significance or modify their text.

Version 2:

Reviewer comments:

Reviewer #2

(Remarks to the Author)

This is a second review of the manuscript entitled "Crystallographic fragment screening supports tool compound discovery and reveals conformational flexibility in human deoxyhypusine synthase". The authors contend that fragment screening identified a number of lead compounds that led them to develop a novel compound (DMP7). Extensive crystallographic data of the structures (Space group $P3(2) 2 1$) obtained by fragment screening is provided for review. DMP7 is located at a crystallographic 2-fold resulting in packing consistent with SG $P3(2)$. Concerns about the crystallographic data for DMP7 were addressed with additional data that will be accessible on publication. Fragment screening is a field in which there is a lot of discussion about the data and what it means; however, I'm satisfied that there is extensive data such that a well-informed reader would be able to make their own conclusions.

Reviewer #1 (Remarks to the Author):

This paper describes an X-ray crystallographic screening of human deoxyhypusine synthase (DHS), using a fragment-based approach. The authors introduce the target and describe the enzymatic mechanism, however it would be good to include a scheme to show exactly what the mechanism is. The authors discuss the possible therapeutic value of targeting DHS and have highlighted several disease areas, specifically they highlight the eIF5A hypusination as a clinically relevant modification. To target DHS the authors have proposed using a fragment-based approach using X-ray crystallography as a screening technique.

The results describe the screening of a 172-compound fragment library however more information needs to be included in the manuscript for example, what is a FragMAXLib, what is the composition and why only 172 compounds? While the size of fragment libraries generally varies, the number of compounds screened is low and generally the smaller the library these tend to be more biased.

FragMAXLib is a composition of low molecular weight compounds designed and assembled at the MAX IV laboratory. As it is continuously curated and has so far only been published on the synchrotron's website, we provided the exact composition used in our campaign in the supplementary materials to ensure reproducibility (Table S1). The size of the library is well suited for crystallographic fragment screening. Comparable libraries may be larger or smaller depending on platform throughput, but they remain orders of magnitude smaller than HTS collections. Nevertheless, fragment libraries are chemically optimized to probe a similar extent of relevant chemical space [Murray, C., Rees, D. The rise of fragment-based drug discovery. *Nature Chem* 1, 187–192 (2009). <https://doi.org/10.1038/nchem.217>; Wollenhaupt, J. et al. F2X-Universal and F2X-Entry: Structurally Diverse Compound Libraries for Crystallographic Fragment Screening.

When these fragments were screened, these were screened at 100 mM concentration and subsequently a significant hit rate was observed. However, one major challenge with the fragment screen was the identification of 37 binding sites, it is not clear as to how you would prioritise these fragments for an elaboration strategy. Have the authors thought of rescreening the fragment hits at lower concentrations of for example 50 mM to 5 mM to see whether the fragment hits can be ranked?

The point of using high concentrations of ligands was to maximize the hit rate; lowering the concentration would inevitably decrease it. We chose to report all identified fragments together with the raw diffraction ensemble used for their identification, as we are aware that different prioritization strategies can be applied by different medicinal chemistry groups. In line with our open science approach, this ensures maximum transparency and reusability of the dataset. Within our study, we clearly prioritized fragments identified in the active site and the strongest binders at the oligomer interface.

Interestingly one of the fragments (VT00065) has been proposed as a covalent fragment. It is not clear how this fragment undergoes covalent attachment; this is an unusual covalent warhead. Have further experiments been carried out for example soaking the fragment VT00065 over prolonged time to examine the timeframe for covalent attachments, this could then be coupled with a mass spectroscopy method to identify covalent attachment. This needs further experiments before this can be classified as a covalent fragment.

The named fragment overlapped directly with the catalytic lysine of DHS, and the only convincing explanation of the observed density was a covalent linkage especially that one of the important step of the reaction is redox-dependent formation of intermediate state on catalytic lysine with imine linkage. Previously we've used the redox mechanism to capture covalently modified K329 forming a transition state analogue. Also the innate redox potential of DHS is known to facilitate a side reaction leading to formation of forming Δ^1 -pyrroline from SPD. The crystals were incubated for a prolonged time (~24 h) in high ligand concentration (100 mM), and under these conditions the event was reproducibly observed in PanDDA maps in both active sites across all tested replicates. This case illustrates that very weak binders often require dedicated detection tools such as PanDDA, since they remain invisible in conventional 2fo–fc maps. We attempted to confirm the linkage by orthogonal method, unfortunately, MS results were too noisy to capture such a low-occupancy modification, and we lacked means to enrich the “linked” protein fraction. For this reason, we relied on the in vitro hypusination assay, which indeed confirmed limited inhibitory activity of VT00065 at very high concentration.

Figures 2 and 3 highlight the structural overlay of the fragments and while they give a good global pictorial overview of the fragment binding sites, it is difficult to see the detail of the fragments binding. While there are four inlaid fragment zoom-ins to their binding sites these remain challenging to interpret, especially where there are multiple overlays of fragment hits. In figure 4 a fragment linking strategy was employed with compound VT00155 to form compound DMP7, it is not clear as to why this fragment was chosen for fragment linking also it is not clear as to what the affinity of the original fragment and elaborated compound was. There are multiple starting points for elaboration and a discussion should be included outlining where these will be.

We prepared illustrative figures for the main text to provide an overview of the results. In addition, we submitted extensive supplementary files, including a PyMOL session with all identified hits overlaid and detailed figures for each instance of binding together with the corresponding maps supporting ligand placement. Fragment VT00155 was chosen for follow-up design because of its location at the oligomeric interface close to the 2-fold axis, which made linking feasible, and because it exhibited strong binding, with clear density visible not only in PanDDA maps but also in conventional crystallographic maps. For clarity we changed a following reasoning in the “Targeting the tetramer interface and design of the follow-up compounds” chapter:

To:

“We decided to explore the potential of fragment-based compound design by linking two molecules present in the close proximity. Many such instances were observed in the vicinity of the interface with pyroquilone (VT00155, fig. 4h) standing out as one of the strongest hits, visible in both PanDDA and conventional maps. “

The results reported in this manuscript are too preliminary to publish in Nature Communications and further structural and biophysical characterisation are needed to ensure that could be considered for publication. A further caveat would be for a fragment-elaboration strategy to be proposed or even to have been explored in greater detail.

We disagree that our results are “too preliminary”. The aim of this work was not to deliver a fully optimized inhibitor, but to establish the structural ligandability of DHS and to demonstrate the potential of fragment-based approaches to probe its regulation. In this sense, the study presents

a complete story at the proof-of-concept level: from target selection, through high-throughput fragment screening, to mechanistic interpretation, in vitro confirmation, and a first example of fragment elaboration at the oligomeric interface.

Further optimization and pharmacological validation will indeed be required to advance towards drug-like molecules. However, such efforts are inherently lengthy and resource-demanding, and fall outside the intended scope of this manuscript. Our contribution is to provide the structural framework and open dataset upon which these future campaigns can be built. Nevertheless we accept the suggestion that the “*Communications Chemistry*” may be better suited for the manuscript and will reach more focused reader group.

Reviewer #2 (Remarks to the Author):

The manuscript entitled "Crystallographic fragment screening supports tool compound discovery and reveals conformational flexibility in human deoxyhypusine synthase" represents a large amount of work. The authors assume a lot of prior knowledge about the protein making it unsuitable for the broader audience of Nature Communications. They leave the reader to do much of the research work rather than making a clear point and then providing evidence in support of it. For example, it is not clear which discussions refer to previously solved structures and which refer to trials included here. More careful thoughts about each figure would make the paper more readable and readers more able to evaluate the work.

We acknowledge that our introduction and discussion may not have provided sufficient context for readers outside the immediate DHS/hypusination field. Our intention was to be concise and to reference both our own and others' previous results, including reviews and commentaries, rather than repeating the detailed background. Nevertheless, we agree that for a broad Nature's Communications Chemistry audience, additional context will improve readability. In the revised manuscript we expanded the introduction to provide a more structured historical background of hypusination and to include a clearer justification of DHS as a target, as well as an overview of prior attempts to develop specific inhibitors. We hope that including this perspective made the rationale and significance of our study clearer for a broad readership.

The authors state that "x-ray crystallographic analysis stands out because it not only provides binary yes/no identification of fragment but also delivers detailed structural information on the fragment's position and orientation within the protein". This implies that the atoms built will be visible in the electron density. However, the inspection of the reports indicate that the fragments they purport to be present are not representative of the molecules modeled. While this is well documented for weakly binding molecules, and alternative methods such as PanDDA have been developed to tease out structural information from low occupancy fragments, this undermines the statement the authors make.

We disagree that the use of PanDDA undermines our statement. On the contrary, it directly supports the point that crystallographic analysis can provide structural information on weakly bound ligands. Conventional crystallographic maps rely on averaging (e.g., symmetry-related reflections, multiple conformations) and can obscure low-occupancy events under noise. PanDDA adds an additional level of statistical averaging across entire datasets, allowing weak but reproducible signals to be extracted with confidence. In our view, PanDDA event maps should be

considered a valid and widely recognized source of structural information (Pearce et al., Nat Commun 2017).

All reported hits in our study are supported by PanDDA event maps of sufficient quality for ligand placement, and these are provided transparently in Supplementary File S6. For stronger binders, the signal is visible not only in PanDDA maps but also in conventional 2fo–fc electron density, which we have highlighted in the figures.

Importantly, PanDDA is now the standard practice in large-scale crystallographic fragment screening campaigns at synchrotron facilities such as Diamond, MAX IV, and SLS. Its use does not diminish the robustness of the analysis but reflects current best practice in the field, ensuring that weak but reproducible fragment binding events are captured and reported consistently.

The authors state that “Unfortunately, the data interpretation was significantly less straightforward than expected for a seemingly homogenous set of structures at avg. 1.7 Å resolution. It appeared that almost all identified PanDDA events were false positives coming either from a partly ordered N-terminus (absent in the apo DHS search model) or predominantly from disordered loops forming an NAD-binding Rossmann fold”. It would have been far better to carefully refine structures and spend time discussing the regions of the protein that became ordered in the absence of NAD. The fact that almost all of the PanDDA events in the absence of NAD were false positives undermines the remaining results. The authors miss an opportunity to discuss in detail the conformational differences that were apparent in the false positives and between apo and NAD bound structures would have been interesting. It would also have given more confidence that they had carefully examined the electron densities and structures, even if this was included only as supplemental data.

The aim of the project was to identify small-molecule binders rather than to analyze DHS flexibility under non-physiological (apo) conditions. Nevertheless, we invested considerable effort in attempting to interpret the apo dataset. We built average models, tested ensemble refinements, performed sub-clustering by unit cell parameters (e.g. BLEND in CCP4), and applied new algorithms provided by the PanDDA2 developers. Despite these efforts, the majority of apparent “hits” overlapped with traces of disordered regions, such as the flexible N-terminus or loops forming the NAD-binding Rossmann fold, and could not be reliably distinguished from noise.

From these attempts, we concluded that reproducible identification of ligand binding requires stabilization of DHS with its natural cofactor NAD. Once stabilized, the system yielded interpretable datasets and reproducible fragment hits. We report these difficulties with the apo form because they illustrate a methodological point of broad relevance: fragment screening of flexible enzymes is significantly improved when natural cofactors are included. We believe this will help others to avoid investing effort in unstable apo systems when a simple experimental adjustment can render the screening more robust.

I found the figures difficult as they were not intuitive and did not illustrate the points that the legend and text suggest. For example, the authors state that they have identified a new conformation of the N-terminal amino acids of one monomer that forms the ball and chain blocking the active site. A schematic showing the different conformations or monomers superimposed would have greatly clarified how the conformations were different. Figure 3A shows the different conformations but again does not state which of the many PDB files it is showing. No electron density is shown for the amino terminal residues which is concerning because the validation files supplied where there residues are built highlight that the density does not fit the atoms as modeled for these residues

(9-21) and this must be investigated and corrected if they wish to make this statement. Figure 3B and C do not help to clarify the conformational change.

Fig. 3 was included precisely to allow comparison between the canonical conformation of the ball-and-chain motif (purple, as indicated in the caption) and the shifted orientation (grey). Insets highlight the interaction sites with labeled residues to clarify how the two conformations differ. To strengthen this point, we provide a dedicated supplementary figure with electron density illustrations. One of the clearest examples are VT00068 and VT00143, where extensive helix-like density was identified not only in PanDDA maps but also in the fo–fc difference density. Similar patterns were observed for other ligands reported in the manuscript. We provide an example with repositioned N-terminus visible in fo-fc map as a Supplementary Figure 10.

WRT the crystallographic data in which the majority of the manuscript focused, I was surprised that so many reflections were flagged as free, especially since their earlier structures had used a more reasonable number (~2000). It may help to take some of the key structures and start again with fewer R-free reflections as they are setting aside many more than necessary. A brief look at some of the files highlighted problems, for example, 9ID8, which contains a covalent inhibitor has B-factor for all atoms of A1I3P of 20 and occupancy of 0.99 in chain A and 0.55 in chain B, so it would appear that these atoms were not properly refined. Other non-physical anomalies can be found in the B-factor refinement, such as the carbonyl oxygen atom having a lower B-factor than the carbon to which it is bonded. While these might seem like minor details, they indicate structures where more careful refinement and study could result in interpretable density. I understand that the goal is to do automated fast screening, however, this doesn't mean that the structures are not carefully inspected.

Free reflections are an undisputedly accepted way to assess the fit of a model to the experimental data [Brünger, 1992]. The discussion of how many reflections should be set aside has been important for the field, with up to 10% originally suggested [Brünger, 1992], and later recommendations converging on ~5% or ~2000 reflections (whichever is lower) as a reasonable compromise. In our previous DHS structures we followed this convention. During the fragment screening campaign, however, it was not feasible to reprocess each crystal individually, and we therefore used the free reflection sets provided by the automated data-processing pipeline. This choice does not materially affect the outcome of the campaign, whose aim was to identify fragment binders across hundreds of datasets rather than to maximize the statistical quality of individual refinements.

The ADP = 20 Å² is a default value set by COOT when placing new atoms such as ligands. During refinement, occupancy may be artificially compensated by inflated B-factors, especially for weakly bound fragments. Since fragment ligands typically bind with low affinity and low occupancy, their refinement on a per-crystal basis is often unstable and can distort the local density. To avoid this, we modeled hits based on the available PanDDA and conventional maps without forcing classical refinement to convergence. This approach is consistent with the original PanDDA workflow, which combines ground-state models with ligand-bound states for robust hit identification.

We are aware of ongoing debate in the community regarding optimal refinement strategies for large-scale screening data [Jaskólski et al., 2022; Weiss et al., 2022]. Our position is that the most efficient approach should be used, even if it is not always the most “elegant” from a single-structure perspective. For large-scale campaigns, treating the ensemble of crystals as a dataset, rather than refining each structure in isolation, represents a pragmatic and widely adopted solution.

Minor note about Supplemental figure 2. It shows the structures of human (apo) and trichomonis vaginalis (NAD present) and seeks to demonstrate that inclusion of NAD resulting in an ordering of residues 284-297. However, the authors do not clearly label which is which, list the PDB codes or include sufficient information in the figure legend to quickly evaluate the figure. The figures are colored according to temperature factor ($\pm 10 \text{ \AA}^2$), with coil thickness indicating higher temperature factors.

We applied due diligence to depict points raised in the text as clearly as possible using static figures, yet there always will remain a small interpretability when inspecting illustrations. We labeled each panel as “apo” or “NAD” and referred to the “left” and “right” panels in the caption, with insets zoomed on the same region of the protein to highlight the presence of NAD.

Both structures presented on Supplementary Figure 2 shows human structures built in the crystals obtained and analyzed by us. The apo structure derives from the 1st CFS campaign in which we used apo DHS and the NAD-bound structure is example of the unbound structure from the 2nd campaign for which NAD was included in crystal preparation. There is no *T. vaginalis* shown here and we are sorry for the confusion.

*Assuming the structure is representative of something physical, the authors do not use any secondary biophysical methods to validate them. The crystal structures are infamous for co-crystallizing with other small molecules, such as ammonium sulfate, PEG, and buffers. This does not imply that they have function. The only non crystallographic data is presented in Figure 4 panels E, J and K. There are several issues with the statistical analysis. The use of SEM is inappropriate; the SD (shows spread of data) should have been used (Krzywinski & Altman; Nature Methods v10 p 922 2013). Using SEM can result in statistical significance where none exists. Actual P-value should have been quoted not $P < 0.05$. Panel j has an undefined * on the side of the green bar. The authors conclude that VT00065 shows does dependent inhibition, but their statistical analysis does not claim this, and inspection of the data presented shows high variability for each concentration. Panel K has a similar issue. In all cases, the authors should include ALL Westerns, not just their representative example. (In addition, they should show the electron density).*

The crystal structure represents one of the possible physical states of the protein, and indeed it is common for crystals, including DHS in our previous studies, to also bind components of the crystallization cocktail. Fragments are inherently weak binders, typically with millimolar affinities, which makes precise affinity determination by classical methods both difficult and of limited immediate value. We view a crystallographic fragment screening (CFS) campaign as a starting point: it establishes structural evidence of ligandability and identifies chemotypes suitable for elaboration. Also the buffer components may be informative for medicinal chemist when

designing new ligands yet neither it, nor individual fragments are expected to “have function” on their own. Literature data reports fragments to be usually a millimolar to high-micromolar binders and as such their K_d is little informative. Reliable binding constants can in turn be measured for optimized follow-up compounds with higher affinity, for example using MST or ITC. Such optimization is ongoing but lies beyond the scope of the current proof-of-concept study.

We included statistical analysis of the Western blot data as a standard practice in data presentation rather than as a detailed statistical dissemination of the data. Per the reviewer's request, we adjusted the plots so that the error bars show SD rather than SEM; however, it is important to note that the choice of error bar does not affect the outcome of the ANOVA analysis used. The majority of the observed variability is inter-assay and inherent to the technique, as we present raw densities normalized only intra-assay for protein loading (using Ponceau S signal). We presented the data as fold change relative to the control, which is common practice in Western blot data presentation to minimize variability. We thank the reviewer for noticing the issue with the plot in panel j, which we corrected. We also included images of all Western blot replicates as supplementary data and report the actual P-values there as well (Fig. S9).

In summary, I do not judge that this manuscript, as written, has provided the data to support their claims and in its current form should be published in a more archival journal. This does not intend to undermine the amount of work done. The manuscript, if further considered, should be completely rewritten to make clear points with simple figures showing high quality data and secondary experiments that validate the hits.

We aimed to provide a detailed description of our efforts to identify small-molecule binders of DHS and to share these results with the scientific community in support of ongoing attempts to develop novel compounds targeting the polyamine pathway. Our focus here is on establishing structural ligandability and providing an open dataset as a foundation; the downstream medicinal chemistry required for full drug development is intentionally beyond the scope of this work. We believe that our work will be of interest to a broad community of *Communications Chemistry* readers.

Reviewer #3 (Remarks to the Author):

In this manuscript “ Crystallographic fragment screening supports tool compound discovery and reveals conformational flexibility in human deoxyhypusine synthase.” Piotr Wilk et al. perform crystallographic fragments screening (CFS) to identify potential DHPS interactors/inhibitors. DHPS is the enzyme that, together with DOHH, catalyzed eIF5A hypusination, a posttranslational modification that is necessary for its activation. eIF5A is a translation factor involved in several pathologies ranging from cancer to diabetes. There is a limited number of hypusination inhibitors, and none are used for the treatment of patients. By screening a FragMAXLib fragment library (172 compounds) they identified 67 unique fragments that bind to DHPS at 136 binding sites. These molecules are classified according to their ability to bind different parts of the DHPS tetramer, the active site, the entrance tunnel, the “ball and chain” motif, the tetramer interface, or the allosteric sites. One molecule (VT00065) seems to bind covalently to DHPS catalytic lysine residue (K329) and could inhibit DHPS. VT00155 binds the DHPS tetramer interface. By linking two VT00065 molecules they obtain a compound (DMP7) that interacts with DHPS and increases its enzymatic activity against eIF5A in vitro. The authors point out that crystallographic fragment could be a useful tool for “probing protein dynamics, identifying novel binding pockets, and investigating

regulatory mechanisms screening”

CFS is a potent technic that allows to identify weak binding compounds that interact with a target protein and provides the molecular mechanism of this interaction. The goal is then to refine the molecule to increase their efficiency. In this manuscript, CSF has been carefully performed and allowed the identification of small molecules that bind different part of DHPS. This data could allow for very interesting further development. However, as such, these data are preliminary since the fragments isolated by CFS are only analyzed by crystallography. Only two molecules are tested for their effect on DHPS enzymatic activity. No convincing inhibitor is identified, and no refinement of the fragment has been performed. So, although the authors state (lines 246-248) “Further optimization of these fragments, particularly to enhance binding affinity and reactivity, could yield a new class of irreversible DHS inhibitors with high specificity and therapeutic potential.” no convincing data are provided indicating that any of the fragment isolated can inhibit DHPS enzymatic activity in vitro, and even less in intact cells or in animals.

Main comments:

The screening hit-rate is 39% with 67 unique fragments. Although CFS allows for the identification of weak-binding organic compounds such a high value question the “realism” of theses interactions. Protein crystallization (lines 424-431) is performed with high concentration of fragments (100 mM), 20 mg/ml of DHPS (around 500µM), in 10% DMSO and PEG. How can the authors be sure that this “weak-binding“ of the fragments with DHPS, and the modification of DHPS observed, can be observed elsewhere than under the crystallization conditions and affect DHPS enzymatic activity?

Fragments identified in crystallographic fragment screening are, by definition, expected to bind only weakly. Their value lies in revealing ligandable sites and chemotypes that can be elaborated, not in providing tight binding in isolation. For ligands with sufficiently strong anticipated binding, orthogonal assays such as MST or ITC, complemented by enzymatic readouts like the hypusination assay, can be applied. Such measurements become meaningful only once affinity reaches a level detectable by these classical techniques.

The authors show that, in the crystallization study, several fragments “induce significant structural rearrangements of crucial regulatory elements” of DHPS (line 27-28). However, only two molecules have been tested for their effect on DHPS enzymatic activity.

We emphasize that demonstrating fragment-induced rearrangements structurally already constitutes an important mechanistic insight. The ability of fragments to shift the conformation of the ball-and-chain motif highlights previously unrecognized regulatory flexibility in DHS. Only two molecules were tested biochemically in this first study, but showing a measurable stabilizing effect in solution would require ligands capable of strong and persistent binding. Achieving this demands substantial medicinal chemistry optimization. While we have not yet pursued this, our dataset establishes the structural basis for such efforts and provides the community with a clear starting point.

VT00065 is presented as an inhibitor. To support this, the authors performed an in vitro assay followed by a deoxy-hypusine Western blot (fig 4e). Although the picture of the Western blot is convincing, the quantification of three experiments reveals an important variability and no significant statistical differences between the control and the treatment with VT00065, even at 20

mM, a high concentration. This questions the ability of this molecule to act as an inhibitor of DHPS. In addition, the authors propose that VT00065 modifies covalently DHPS catalytic lysine K329. If such reaction happened in the in vitro experiment, one could expect a potent inhibition of DHPS activity, and again, questions the hypothesis that the data obtained in the crystallization study can be reproduced in a classical enzymatic reaction buffer, using lower concentration of fragments.

The inhibitory effect of VT00065 was observed only at high concentrations, consistent with its weak occupancy in the crystals, where it was detected in PanDDA event maps but not in conventional 2fo–fc density. We interpret this as reflecting low saturation of the active site. Our ongoing efforts focus on designing derivatives with tighter binding to DHS, which would allow the redox reaction to proceed more efficiently and shift the equilibrium toward product formation. DHS is also known to catalyze a side reaction leading to cyclization of the polyamine substrate into Δ^1 -pyrroline, a process likely requiring NAD as a redox partner. VT00065 appears to react in a manner similar to this side reaction and our structural observations provide the foundation for developing optimized analogues with improved activity. Of note, in this assay we do not incubate DHS with VT00065, as at the very low concentration required for the assay (50 nM), DHS is not sufficiently stable. Therefore, given the low affinity of this compound, it would be unlikely to observe “a potent inhibition of DHS activity” under these conditions.

To convince the readers, the author should refine at least one fragment to show that VT00065 could be used as a building block to develop efficient DHPS inhibitors in vitro. These molecules could then be tested in intact cells to determine to what extent the data obtained here could be used for the development of DHPS inhibitors.

We have already explored several follow-up designs based on VT00065, but so far none have shown improved binding compared to the initial hit. Optimization of covalent fragment binders is inherently challenging and, as is well recognized even in large pharmaceutical laboratories, can require many years of iterative medicinal chemistry. While we cannot predict the timeline for achieving a significantly improved analogue, this remains an active direction in our group, and the present study provides the structural basis upon which such future optimization efforts can build.

The discovery of DMP7, that activates DHPS in vitro, is particularly interesting. The author could test if this molecule activates eIF5a hypusination in intact cells. This raised also the possibility that the experimental conditions in the in vitro DHPS activity assay are not optimal. The authors could test, in vitro, if VT00065 is more efficient to inhibit DHPS when it is “fully” active, in presence of DMP7.

In this study we limited our analysis to in vitro assays using purified components, as the available fragments were not expected to show measurable activity in intact cells. We fully agree that extending the methodology to cellular systems will be an important next step, and we are prepared to do so once ligands with sufficient in vitro potency become available.

Overall, we view this study as a proof-of-concept resource rather than a completed drug discovery project. By structurally mapping DHS ligandability, demonstrating fragment-induced conformational changes, and validating representative hits biochemically, we provide the

foundation on which future medicinal chemistry and cellular studies can build. This dataset expands the scope of fragment screening beyond conventional inhibitor discovery and offers the community a unique starting point for developing DHS-targeted chemical tools and therapeutics.

We believe that DMP7 does not activate DHS in vitro but rather stabilizes its structure, leading to increased stability at the very low concentration required for the assay (50 nM).

Minor comments

- In fig 4e, 4j and 4k the authors should provide, in the same figure, a Western blot for DHPS and eIF5A to verify equal loading for each condition.

The loading control were included at the Fig. 4.

- *Line 366-367 “Modifications aimed at increasing affinity, such as merging VT00065 with GC7, could enhance AS pocket saturation, improving the yield of covalent DHS inactivation.”. One could object that the IC50 of GC7 for DHPS is already high (17 nM, PMID: 8514754). Moreover, GC7 is a competitive inhibitor, and as stated by the authors (80-81) “GC7 affects multiple pathways beyond hypusination, making it difficult to isolate its specific role. ». So, the benefit of such chemical modification does not seem obvious.*

The rationale for merging VT00065 with GC7 is to combine the advantages of both molecules: the higher affinity of GC7 with the reaction-based covalent specificity of VT00065. Such a design aims to retain strong binding to DHS while improving selectivity through the unique covalent mechanism.

Reviewer #2 (Remarks to the Author):

The manuscript submitted to Communications Chemistry is a revised manuscript and contains the authors' rebuttals to the earlier reviewers' comments. The manuscript describes the results of screening human deoxyhypusine synthase against a library of 172 compounds using a fragment screening approach to identify the locations of potential inhibitors that might be lead compounds with future therapeutic value. The intended audience is for researchers in all areas of chemical sciences.

In response to the original comments, a rewriting of the introduction is appreciated and improves the readability of the manuscript, especially as it is being targeted to a community who are potentially less knowledgeable about biochemistry. Reviewer 1 suggested a mechanistic scheme, and given the intended audience, I think this would be a valuable addition, especially since there is reported to be a covalent inhibitor. The rebuttal addresses the concerns about the change in conformation of the N-terminal helix - the inclusion of the Fo-Fc electron density map for the N-terminus in supplementary figure 10 is noted.

Reply. *We appreciate recognition of our modifications to the original text. Following Reviewer's advice we have included an overall scheme of the reaction catalyzed by DHS as a supplementary Figure S1. Please also note that we also referenced previous description of DHS catalytic cycle including our own work in Nat. Comm. as well as Afanador's et al. in Structure (ref. 50).*

The development of the DPM7 molecule is one of the key findings. I looked carefully at the data collection statistics and structure quality report as the electron density looked a little less well ordered than I might have expected. I noticed that it diffracts to lower resolution than the other datasets and that it is merohedrally twinned. The overall R_{merge} is 21% and R_{meas} is 22%, suggesting that the twinning may not have been handled in refinement. The experimental methods discuss the reason for the lower symmetry space group, but do not specifically state that the twinning was taken into consideration. I apologize for not noticing this in the initial review. However, the authors should address this either in the data analysis section of materials and methods, explaining why the values are so high if they did account for it, or re-refine the structure appropriately if they didn't.

Reply:

We appreciate this insightful observation; however, the elevated R_{merge}/R_{meas} values reflect lower overall crystal quality rather than true merohedral twinning. Such increases can arise from differences in crystal harvesting, crystallization batch variability, use of a different X-ray source, or other random factors affecting diffraction behaviour. Importantly, Phenix.xtriage analysis does not indicate twinning:

Phenix.xtriage

```
<I2>/    2.111 (untwinned 2.000, perfect twin 1.500)
<F>2/<F2>  0.770 (untwinned 0.785, perfect twin 0.885)
<|E2-1|>    0.763 (untwinned 0.736, perfect twin 0.541)
```

If anything, these values more closely resemble the effects of pseudotranslation rather than merohedral twin domains. We therefore disagree that a twin operator should be applied during refinement. The lowered symmetry and near-equivalence of certain reflections can mimic some

statistical signatures of twinning, but in this case the apparent behaviour arises from intentional symmetry reduction rather than genuine twin fractions.

As noted in the Methods section, the ligand locally breaks the parent $P3_221$ symmetry. Refinement in the higher-symmetry space group would artefactually force the ligand and adjacent linker atoms into overlapping positions, producing smeared or ambiguous density. Refinement in a lower-symmetry setting correctly accommodates these locally non-equivalent positions. Because the underlying lattice retains near-pseudosymmetry, reflections that would normally be symmetry equivalents appear as separate measurements with similar intensities, thereby elevating R_{merge}/R_{meas} without indicating twinning.

Finally, despite the higher merging statistics, the refinement yielded R_{work}/R_{free} values of 17.45% / 19.72%, which are entirely consistent with high-quality models of comparable resolution.

In response to the rebuttal addressing PanDDA. I understand what the process is intended to resolve and am aware of the excellent work of Frank von Delft and his group. I would refer the authors to Pearce et al Acta Cryst D Struc. Biol (2017) vol73 pg 256-266, Gahbauer et al, PNAS v120 e2212931120 and the recent paper in Nature Communications (2025) v16 p8930. In the Acta Cryst and Nat. Comms papers, von Delft and his team provide great examples of fragment screening and the use of PanDDA to screen a 1076 fragment library. The hits they identify are more in line with the 20% or so typical hit rate. The PanDDA event maps in the papers show credible density for their compounds. Further, inspection of the structure reports for the compounds they identified show clear and appropriate $2F_o-dF_c$ electron density maps and little, if any, mF_o-DF_c density contoured at 3σ . This indicates that the molecules are present. By comparison, the same cannot be seen in event maps included in supplemental figure 5 for several of the compounds, for example – VT00219, VT00221, VT00228, VT00229, VT00257, VT00405, VT00428.

Reply: *We acknowledge that several ligands exhibit weaker electron density than typically expected for high-occupancy fragment binders. However, all reported hits were first identified using the PanDDA algorithm, which applies rigorous statistical thresholds to detect low-occupancy ligand states. Each candidate was subsequently examined by an experienced curator across multiple contour levels and map types ($2F_o-F_c$, F_o-F_c , and PanDDA event maps) to ensure that the signal was reproducible and interpretable. Importantly, the complete dataset, including all PanDDA outputs, ground-state models, and duplicate soaks, is publicly available on Zenodo, allowing independent verification. The weaker densities observed for certain fragments are consistent with low-occupancy binding events commonly encountered in crystallographic fragment screening and do not preclude their validity when supported by PanDDA statistics and manual inspection.*

All reviewers have raised concerns about the number of hits identified, characterization of VT00065 as a covalent inhibitor and the realism of the interactions. The authors rebut the reviewers' comments stating that their work is a proof of concept. In addition, they disclose that they are co-inventors of the compound, making publication important. Providing the question about twinning is resolved, there is sufficient supplemental data for an informed reader who is familiar with fragment screening to evaluate the data for themselves.

Reviewer #3 (Remarks to the Author):

The manuscript titled "Crystallographic fragment screening supports tool compound discovery and reveals conformational flexibility in human deoxyhypusine synthase" by Piotr Wilk et al. was

previously submitted to Nature Com. As a reviewer I found this study particularly interesting. I had a few questions and comments, most of which have been addressed by the authors in their rebuttal letter.

The manuscript presents an engaging narrative on crystallographic fragment screening used to identify new inhibitors of DHPS. It also suggests potential new regulatory mechanisms by providing insight into the modification of DHPS conformation and it uncovers binding pockets. This work constitutes a proof-of-concept for a new method aimed at discovering hypusination inhibitors which, after further optimization, could potentially be used for treating various pathologies associated with eIF5A dysregulation such as cancer. However, there is still one important issue that needs to be addressed. In Fig 4e the authors performed an in vitro hypusination assay with increasing doses of VT0065. While the Western blot presented is convincing, the error bars between control condition and the highest inhibitor concentration overlap. This is confirmed by the p values provided in this new version of the manuscript (fig S9). Indeed, the p value between the ctrl and VT0065 is 0.25 which is usually not considered statistically significant (the commonly accepted threshold being < 0.05). Thus, although the authors state that: “ In this assay, VT0065 exhibits low millimolar inhibitory properties” the statistical analysis does not support the fact that an inhibitory activity, even low, is demonstrated by these experiments.

To resolve this, the authors should either increase the number of experiments to reach statistical significance or modify their text.

Reply: *We recognize the Reviewer’s cautious interpretation and we have moderated our claim and modified the paragraph of concern. Now it reads “In line with the crystallographic observation are in vitro analyses of the rate of hypusination in the presence of VT0065 assessed by western blot using a deoxyhypusine-specific antibody. In this assay, VT0065 **shows some signs of inhibitory properties** (Fig. 4e).”*